# Maternal Tryptophan Catabolites and Insulin Resistance Parameters in Preeclampsia

**DOI:** 10.3390/biom13101447

**Published:** 2023-09-26

**Authors:** Zainab Abdulameer Jasim, Hussein Kadhem Al-Hakeim, Samaneh Zolghadri, Agata Stanek

**Affiliations:** 1Department of Biochemistry, Shiraz Branch, Islamic Azad University, Shiraz 7198774731, Iran; zainab23490@gmail.com; 2Department of Chemistry, Faculty of Science, University of Kufa, Najaf 54001, Iraq; headm2010@yahoo.com; 3Department of Biology, Jahrom Branch, Islamic Azad University, Jahrom 7414785318, Iran; 4Department and Clinic of Internal Medicine, Angiology and Physical Medicine, Faculty of Medical Sciences in Zabrze, Medical University of Silesia, Batorego 15 St., 41-902 Bytom, Poland

**Keywords:** preeclampsia, insulin resistance, tryptophan catabolites, kynurenic pathway

## Abstract

Preeclampsia (PE) is a pregnancy-related disorder characterized by high blood pressure and proteinuria in the third trimester. The disease is associated with many metabolic and biochemical changes. There is a need for new biomarkers for diagnosis and follow-up. The present study examined the diagnostic ability of tryptophan catabolites (TRYCATs) and insulin resistance (IR) parameters in women with PE. This case-control study recruited sixty women with preeclampsia and 60 healthy pregnant women as a control group. Serum levels of TRYCATs (tryptophan, kynurenic acid, kynurenine, and 3-hydroxykynurenine) and IR parameters (insulin and glucose) were measured by ELISA and spectrophotometric methods. The results showed that PE women have a significantly lower tryptophan level than healthy pregnant women. However, there was a significant increase in kynurenic acid, kynurenic acid/kynurenine, kynurenine/tryptophan, and 3-hydroxykynurenine levels. PE women also have a state of IR. The correlation study indicated various correlations of IR and TRYCATs with clinical data and between each other, reflecting the role of these parameters in the pathophysiology of PE. The ROC study showed that the presence of IR state, reduced tryptophan, and increased 3-HK predicted PE disease in a suspected woman with moderate sensitivities and specificities. In conclusion, the pathophysiology of PE involves a state of IR and an alteration of the TRYCAT system. These changes should be taken into consideration when PE is diagnosed or treated.

## 1. Introduction

Preeclampsia (PE) is a systemic disorder of pregnancy, affecting 2–8% of pregnancies around the world [1,2], characterized by proteinuria and hypertension in the third trimester [3]. However, in some cases, the onset of PE may occur as early as the second trimester. PE can develop as early as the second trimester. It is associated with widespread endothelial damage and future cardiovascular disease [4]. The consequences of developing this disease can be severe, leading to maternal and newborn complications and deaths, including placental abruption, preterm birth, fetal growth restriction, stillbirth, and maternal death [5]. The pathogenesis of PE is believed to result from oxidative stress-induced placental injury or hypoxia, leading to the release of a maternal systemic anti-angiogenic imbalance [6].

Women with PE have worse metabolic and biochemical profiles than those without PE [7]. In PE as a metabolic disorder, many pathways are altered, especially insulin resistance (IR) and the kynurenine (KP) pathway. One of the metabolic abnormalities associated with PE is IR, which some researchers have considered a condition of PE [8] because IR is more common in women predisposed to develop PE months before clinically apparent disease [9]. Reduced tissue blood flow may contribute to the increased IR seen in PE [10]. Mid-trimester maternal IR is associated with later PE [11]. Hepatic and metabolic profiles, especially those associated with IR, are higher in PE than in healthy pregnant women [12]. One of the less studied pathways in the PE is the kynurenine pathway (KP), which degrades 95% of the tryptophan (TRP) to kynurenine (KYN) and either its neurotoxic or neuroprotective immunogenic metabolites [13]. KYN is converted to kynurenic acid (KYNA) by transamination to 3-hydroxykynurenine (3-HK) by hydroxylation. In subsequent steps, it is non-enzymatically converted to 3-hydroxyanthranilic acids (3-HA), forming quinolinic acid (QA) and picolinic acid (PA) [14]. Decreased TRP catabolism leads to a peripheral increase in serotonin, which may be a key marker of nutritional status and an effector promoting lipid absorption and storage [15]. Reduction of plasma TRP levels and an elevated KYN to TRP ratio are significant indicators of an adverse clinical prognosis [16]. Researchers have observed decreasing TRP and increasing KYN/TRP ratios (an index of TRP degradation and IDO-1 activity) across pregnancy, along with evidence of a role for inflammation in these effects [17]. PE women have a plasma higher KYN/TRP ratio, an index of enzyme activity, than non-pregnant women, but lower than pregnant women [18]. During pregnancy, the placenta produces an increased amount of KYN from TRP [19]. KP plays a crucial role in immunomodulation and protection. Elevated levels of TRYCATs can lead to significant immunosuppression or cytotoxicity [20]. The augmentation of KP activity has the potential to present a novel therapeutic approach for the management of PE [21].

A possible link between IR and KP molecules is that serotonin induces pancreatic insulin release, adipogenesis, and lipogenesis [22]. Serotonin stimulates efficient lipid storage, consistent with its evolutionarily conserved role as an essential modulator of energy balance [15]. Hyperserotonemia in PE promotes inflammation through immune cells, cytokines, and metabolic-based mechanisms. These immune mechanisms can be applied to treat platelet and vascular endo-phenotypes in PE [23]. The present study investigated the correlation between these two pathways, IR and KP, in PE women free from overt inflammation.

## 2. Subjects and Methods

### 2.1. Subjects

#### 2.1.1. Patients

This case-control study was conducted at the Women’s Obstetrics and Gynecology Hospital in Karbala Governorate, Iraq, from April to May 2023. Sixty patients with PE have participated in this study. The criteria of the American College of Obstetricians and Gynecologists were used for the diagnosis of definite PE [24]. Pregnant women were considered PE if they had a systolic blood pressure of 140 mm Hg or diastolic blood pressure of 90 mm Hg after 20 weeks of gestation and proteinuria (They should have normal blood pressure before). In the present study, the patients followed these criteria, and all patients had positive proteinuria in the dipstick test. They were fasted overnight and treated with methyldopa (Aldomet^®^). The gestational age was calculated from the last regular menstrual period, and the fundal height and ultrasound results were used for those women who could not remember their last menstrual period. Gravidity was recorded as the total number of pregnancies, including abortion, ectopic pregnancy, and any other pregnancies documented on the chart. Parity is defined as the number of deliveries after 28 weeks of gestation, including intrauterine fetal demise (IUFD) and stillbirth. The study was approved by the Institutional Review Board (IRB) of the Training and Human Development Center, Kerbala Health Department (Document number 585/2023), Kerbala, Iraq, which complies with the International Guideline for Human Research standards mandated by the Declaration of Helsinki.

#### 2.1.2. Controls

Thirty pregnant women with no apparent abnormalities and a gestational age of >20 weeks were selected as a control group. These subjects were stable and free from emotional and physical problems and normal blood pressure (around 120/80 mmHg).

#### 2.1.3. Exclusion Criteria

Patients with systemic diseases, especially diabetes, heart disease, viral hepatitis, and kidney disease, were excluded from the study. All patients with positive C-reactive protein (CRP) were also excluded to rule out visible inflammatory conditions. Furthermore, women with serum FBG > 25 mM and fasting insulin > 413.38 pM (57.6 mIU/L) based on the HOMA calculation, and patients with evident major diabetic complications (such as heart disease, liver disease, and renal diseases), were excluded from this study. We also excluded patients who were receiving metformin based on previously mentioned facts [25] due to the well-known effect of metformin on IR [26] and insulin sensitivity [27].

### 2.2. Methods

#### 2.2.1. Biomarkers Assays

Fasting venous blood was collected from all participants shortly after immediately after the hospital admission. Since tryptophane is an amino acid found in plants and proteins, the food intake increases the level of TRP and, subsequently, its metabolites. It is also recommended to measure glucose and insulin in the fasting state because the sugar in food increases glucose and insulin levels. The blood was allowed to clot for 10 min at room temperature, then centrifuged at 1006× *g* for 10 min. The serum was separated, transferred to Eppendorf tubes, and stored at −80 °C for further analysis. Serum levels of insulin, KYN, KYNA, TRP, and 3-HK were measured using commercial ELISA kits (Nanjing Pars Biochem Co., Ltd., Nanjing, China). These kits were established on a sandwich approach and provided inter-assay coefficients of variation (CV) of less than 10%. The glucose level in the serum was tested using ready-for-use kits (Spinreact^®^, Barcelona, Spain) spectrophotometrically. The homeostasis Model Assessment 2 (HOMA2) Calculator© (Diabetes Trials Unit, University of Oxford; https://www.dtu.ox.ac.uk/homacalculator/download.php, accessed on 15 May 2023) was used to estimate insulin sensitivity (HOMA%S), β-cell function (HOMA%B), and IR (HOMA2-IR) from fasting glucose levels and serum insulin. Additionally, we utilized the CRP latex slide test (Spinreact^®^, Barcelona, Spain) to assay CRP serum levels.

#### 2.2.2. Statistical Analysis

Statistical distribution is divided into two types: non-parametric and normally distributed variables. In this study, the Lilliefors corrected Kolmogorov-Smirnov test was applied to examine the statistical distribution of the results. Then, the normally distributed variables were expressed as mean ± standard deviation. Analysis of Variance (ANOVA) was used to compare the groups in the measured parameters. The correlation between parameters was estimated by calculating Pearson’s correlation coefficients. The results were expressed for the non-parametric variables like medians and (25–75% interquartile). Mann-Whitney U test was used to compare non-normally distributed variable between groups. The correlation between parameters is estimated by calculating Spearman’s coefficients (ρ, rho). The distinction among groups is considered different statistically when *p* < 0.05. Receiver operating characteristics (ROC) curves were measured to examine the diagnostic ability of the measured biomarkers to diagnose disease and abortion. The cut-off values of the concentrations produce the best sensitivity and specificity from the area under the curve (AUC). Confidence intervals of the AUCs were also calculated to determine the precision of the calculated AUC (the narrower interval indicates a more confident conclusion). Youden’s J statistics were measured to indicate the direction of the change in AUC (a positive value means the biomarker increases with diagnosis, while a negative value indicates that the biomarker decreases with diagnosis). The cut-off values were chosen as the concentration corresponding to the highest Youden’s J statistic value. SPSS Statistics version 25 and IBM-USA performed all statistical analyses. The figures were structured by using Microsoft Office Excel 2021.

## 3. Results

### 3.1. Comparison of Demographic and Clinical Data between PE and Control Group

The results of the demographic and clinical data of the PE and control groups are presented in Table 1. The results did not show significant differences between the groups in age, residency, nullipara, multipara, gestational age, parity, gravity, and number of children. There was a significant increase in the SBP, DBP, family history, previous abortion, and Caesarean deliveries in PE patients compared to the control group.

### 3.2. Comparison of the TRYCATs between PE and Control Group

The comparison between PE patients and control groups in serum TRP level is presented in Table 2. The results showed a significant decrease in serum TRP between PE patients in comparison with the control groups. While there is a significant increase in serum KYNA, 3-HK, KYNA/KYN, and KYN/TRP in PE patients compared with the control groups. The results also showed no significant difference in serum KYN between PE patients and control groups.

### 3.3. Comparison of IR Parameters between the PE and Control Group

The results of IR parameters in the PE patients and control groups are presented in Table 3. The results showed significant increases in fasting serum glucose and insulin, HOMA2IR, and I/G ratio in the PE patients compared with the control group. The HOMA%S of PE patients was significantly lower than that of the control group. However, no significant differences were observed in HOMA%B between the groups.

### 3.4. Intercorrelation between TRYCATs and Other Parameters

The correlation coefficients between the biomarkers measured and the sociodemographic and clinical data are presented in Table 4. The results showed significant correlations between TRP and gravidity, number of parity, and the presence of higher age of onset. There was a significant inverse correlation between TRP, SBP, and DBP. Serum 3-HK had a significant correlation with the SDP. While KYN, KYNA, and their ratio KYNA/KYN did not have a significant correlation with the sociodemographic and clinical data of the subjects studied. Also, the serum KYNA/KYN ratio had a significant correlation with HOMA2 IR and was inversely correlated with the HOMA%S. While TRP, 3-HK, KYN, KYNA, and KYN/TRP did not have a significant correlation with the insulin resistance parameters of the subjects under study.

### 3.5. ROC of the IR Parameters

Receiver operating characteristics (ROC) analysis was performed to determine the diagnostic sensitivity and specificity of the measured biomarkers for diagnosing the PE subjects. The ROC curves are plotted in Figure 1. The coordinates of the ROC results and the concentration cutoff that produces the best sensitivities and specificities are presented in Table 5. Based on the results, increases in I/G, HOMA2IR, insulin, and glucose higher than the estimated cut-off values indicated that subjects could have PE with significant sensitivities and specificities (*p* < 0.05). Furthermore, the decrease in HOMA2%S lower than the cut-off value (94.25%) revealed that the subjects may have PE in sensitivity and specificity of 63.3%. While HOMA%B did not have significant predictive values (*p* = 0.742).

### 3.6. ROC of the TRYCATs Parameters

The ROC analysis was carried out to determine the sensitivity and specificity of the biomarkers detected by TRYCATs in the diagnosis of patients with PE. The ROC curves are shown in Figure 2. Table 6 shows the ROC coordinates, as well as the cut-off values that produce optimal sensitivities and specificities. The decrease in TRP was lower than the cut-off value (9.169 ng/mL), indicating that the subjects may have PE disease with a sensitivity of 66.7% and a specificity of 68.3%, respectively. The results showed in Table 6 that the increase in 3-HK higher than the cut-off value (3.849 ng/mL) indicates that the subjects may have PE with a sensitivity of 65.0% and a specificity of 66.7%. KYN/TRP higher than the cut-off value (0.370) indicates that the subjects may have PE with a sensitivity of 65.0% and a specificity of 64.3%. The increase in the KYNA/KYN ratio higher than the cut-off value (23.451 ng/mL) indicates that the subjects may have PE patients with a sensitivity and a specificity of 60.0%. Serum KYNA level higher than the cut-off value (78.678 pg/mL) indicates that the subjects may have PE patients with a sensitivity and a specificity of 60.0%, respectively.

## 4. Discussion

The first important finding of the present study is the alteration in the KP metabolites of PE women compared to healthy pregnant women resulting in reduced TRP and increased other catabolites, suggesting that placental defects may be present in PE women. The KP plays an essential role as a local signaling pathway, providing a source of de novo NAD^+^ synthesis and regulating the supply of TRP and KYN metabolites to the growing fetus [28]. The placentas of women diagnosed with PE exhibited decreased TRP levels compared to healthy women [29]. The metabolic signal associated with tryptophan metabolism in the model across gestation may be connected to preeclampsia’s immunological signature, emphasizing the relevance of immune changes in later stages. Furthermore, the activity of the KP increases under inflammatory conditions, and the different metabolites produced by this pathway can exert potent immunoregulatory functions [29]. The placenta also plays a significant role in fetal serotonin production, and any disturbances in its synthesis within the placenta can have implications for fetal forebrain development [30]. Previous research has demonstrated an intriguing finding that serotonin levels in the placenta are elevated in women who experience PE during pregnancy; however, the precise impact of this increase has not yet been fully understood [31,32]. Some researchers found no significant difference in serum TRP and KYN in PE women compared with the controls [33]. These results are due to the methods and sample size heterogeneity. While, Zhao et al., (2022) found an increase in maternal TRP levels compared with normotensive women [34]. This unusual result is due to the PE group in their study are not BMI-controlled; where the BMI is significantly different between PE and the control normotensive group. Increased weight or obesity are known factors affecting the results of TRP and KYN/TRP ratio [35]. The intermediates of the KP of TRP degradation (KYNA and QA) recreate opposing functions in inflammatory conditions [36]. Increased levels of ROS, IFN-γ, IL-6, and IL-1β may generate indoleamine-2,3-dioxygenase-1 (IDO-1) during disease, which starts TRP catabolism, thus decreasing serum TRP and increasing TRYCATs, including KA, KYN, 3-3HK, QA, and xanthurenic acid (XA). The TRYCAT pathway activation defends against hyperinflammation by different processes, including TRP starvation, scavenging ROS, and adverse immunoregulatory effects [37]. Similarly, some TRYCATs, such as KA and XA, have antioxidant effects [38]. Moreover, 3-3HK, KYN, XA, KA, and QA have adverse immune regulatory roles, such as inhibiting IFN-γ production [39]. However, following TRYCATs overproduction, several detrimental consequences can appear, including immune activation, oxidative stress, and neurotoxic effects [40]. Furthermore, TRP-metabolism was activated in treatment-naïve pulmonary arterial hypertension (PAH) patients, likely mediated through IL-6/IL-6Rα signaling. Additionally, KP metabolites predict the reaction to PAH therapy and the survival of PAH patients [41]. This process may link the TRYCATs with hypertension in PE.

Maintaining a continuous increase in TRP availability is an unachievable idea and can harm both the mother and fetus, with long-lasting and irreversible effects [42]. Placentas of women with PE exhibited a decreased content of TRP and an elevated KYN/TRP ratio; however, no statistically significant alterations were observed in the downstream metabolites [43]. Furthermore, excessive TRP metabolism may induce pathological immunosuppression through overproduction of TRYCATs [44,45]. It is thought that the augmented uptake of TRP by placental trophoblasts may serve as a compensatory mechanism in response to the decreased expression of IDO-1 observed in PE placentas [46]. A comparative analysis of the placentas revealed dysregulation of TRP metabolism in the placentas of women with PE, compared to the controls [43]. Furthermore, placental TRP concentration was found to be elevated in early-onset preeclampsia while reduced in late-onset PE [47]. The concentration of placental TRP was found to be elevated in cases of early-onset preeclampsia, whereas it was observed to be reduced in cases of late-onset preeclampsia [47]. These results may explain some controversy in the findings of TRP in PE women. Another explanation of the controversy in TRP results in PE is possibly due to methodological differences [48].

During pregnancy, there is an increase in KYN production by the placenta, while the contribution of fetal organs remains relatively low throughout the gestational period [19]. Therefore, it can be inferred that both pregnant groups, the PE and the control, exhibit similar changes in serum KYN levels through a shared mechanism. The transformation from TRP to KYN may be considered one of the alternative pathways for TRP metabolism, triggered by specific immune factors that occur during pregnancy. During pregnancy, the placenta produces an increased amount of KYN from TRP [19]. In PE, it appears that this process is increased more in PE and makes KYN elevated than in normotensive pregnant women. The presence of IDO-1, a crucial enzyme involved in KP induction, has been identified in the placenta [49]. A reduction in plasma TRP levels and an elevated KYN to TRP ratio are significant indicators of an adverse clinical prognosis [16]. During subsequent stages, a nonenzymatic conversion occurs, resulting in the transformation of the compound into 3-HA, which subsequently forms QA and picolinic acid (PA) [49,50]. The TRYCAT pathway serves as the primary route for the catabolism of TRP. In cases of excessive activity, this pathway can result in TRP depletion and the generation of neuroactive metabolites, including KYN, KYNA, 3-HK, anthranilic acid, 3-HA, XA, QA, and PA. These metabolites exhibit various functions, including neurotoxic and neuroprotective effect [37,51]. Certain TRYCATs, namely 3-HA, 3-HK, and QA, induce neuro-oxidative toxicity characterized by oxidative cellular injury and lipid peroxidation [37,40,52].

The KYNA concentration was found to increase during the initial trimester of pregnancy in women diagnosed with PE. Moreover, a positive correlation was observed between KYNA and picolinic acid levels and the presence of proteinuria in women diagnosed with preeclampsia during the third trimester of gestation [47]. Therefore, it is plausible to hypothesize whether increased levels of KYNA serve as a compensatory mechanism to mitigate crucial pathological characteristics in PE [40,53]. Placentas obtained from women with PE exhibited an elevated KYN/TRP ratio, indicating an increase in the production of KYN relative to TRP. However, no significant alterations were observed in downstream metabolites. These findings can be attributed to a decrease in the expression of IDO-1, an enzyme responsible for the conversion of TRP to KYN. Furthermore, compensatory upregulation of TDO expression, which is involved in the same metabolic pathway, was detected in the placentas of women with PE [43].

The second important finding of the present study is the correlation between IR parameters and TRYCATs in PE women. The KP parameters have been found to be strongly linked to preeclampsia throughout pregnancy [54]. KYNA functions as a glutamate receptor antagonist, thereby inhibiting the secretion of insulin from pancreatic β-cells [55]. The observed effects of KYNA are fundamentally contrary to the symptoms commonly observed in patients with PE, a condition distinguished by increased IR and hyperlipidemia [56]. KP can be upregulated by both chronic low-grade inflammation and stress, both of which are involved in the development of IR and T2DM [57], and potentially T1DM [57]. PE women exhibited higher serum glucose, insulin, and the HOMA-IR index in their later years than women with normotensive pregnancies [7]. Therefore, dysregulation of TRP metabolism is believed to be a significant factor in the development of T2DM [58].

The third significant finding is that the diagnostic properties of the TRYCATs and IR for PE in suspected women are moderate. Table 5 demonstrates that the decrease in TRP levels and the increase in 3-HK levels are the most effective indicators for diagnosing PE in women suspected to have the condition. However, the sensitivities and specificities exhibit a moderate level. Previous studies have reported a decrease in IDO-1 expression and the activity of TRP degradation in the placenta during PE. Additionally, a correlation has been observed between the reduced activity of TRP degradation in the placenta and the severity of the disease [59]. Furthermore, an association is found between the ratio of TRP/KYN and the duration of gestation [42]. The findings presented in Table 6 demonstrate that the presence of an IR state can be considered an acceptable predictive factor for PE in women suspected of having the condition. Nevertheless, the sensitivities and specificities, which are approximately in the sixties, suggest moderate significance in the diagnosis of PE. In a previous investigation, IR has been identified as a potential biomarker for diagnosing PE [60]. On the other hand, individuals with PE often exhibit pronounced IR [61], suggesting that IR is a contributing factor in the pathogenesis of PE [62]. The timing of PE onset is influenced by mechanisms related to pathological insulin resistance, hyperinsulinemia, and changes in the fetal-placental unit [63]. IR constitutes a significant component of metabolic syndrome and serves as a prognostic indicator of the development of cardiovascular disease in the later stages of life. Women predisposed to develop preeclampsia have been observed to exhibit elevated levels of IR several months before the manifestation of clinically evident symptoms [9]. The increase in insulin resistance observed in preeclampsia may be influenced by a decrease in blood flow to tissues [10]. However, many other factors also affect the levels of IR parameters rather than the PE alone, leading to lower sensitivities and specificities.

## 5. Limitations of the Study

The most important limitation of this research is the relatively small number of patients, which does not allow for modeling of many metabolites at once (or other clinical factors), which could improve the predictive ability. Also, the high cost prevented us from measuring the enzymes of the kynurenine pathway, such as IDO-1, IDO-2, and TDO, as measuring these enzymes provides more valuable information to interpret the results of this research. Another limitation is that we cannot determine the specific cause-effect relationship between the disease and the measured biomarkers.

## 6. Conclusions

From the outcomes of the current investigation, the PE women have a considerably lower TRP level and higher KYNA, KYNA/KYN, KYN/TRP, and 3-HK levels than the healthy pregnant women. Also, the PE women have a state of IR. Correlation research demonstrated numerous associations of IR and TRYCATs with clinical data and each other, demonstrating the importance of these parameters in the pathophysiology of PE. The ROC analysis showed that the presence of an IR state was a predictor of PE illness in suspected women with modest sensitivities and specificities. Also, ROC research suggested that the decrease in TRP and rise in 3-HK are the best variables for PE diagnosis in a suspected woman. However, the sensitivities and specificities are modest.

## Figures and Tables

**Figure 1 biomolecules-13-01447-f001:**
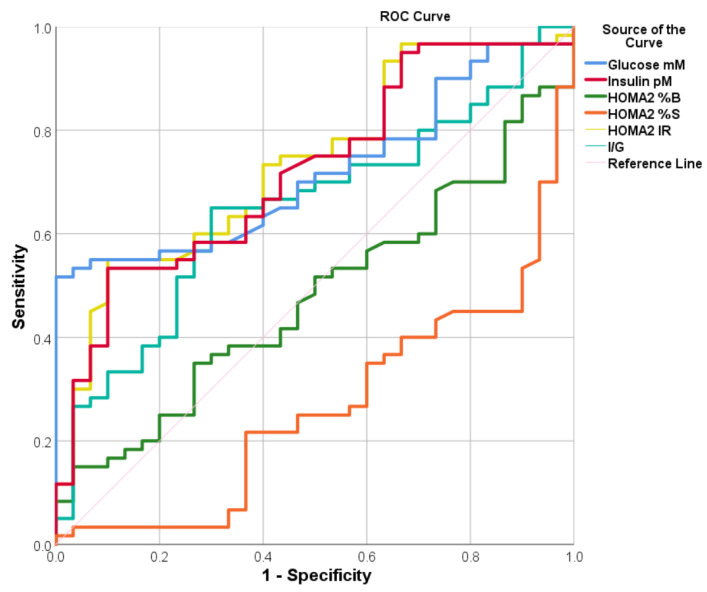
Receiver operating characteristic curves (ROC) of the insulin resistance parameters used for the diagnosis of PE.

**Figure 2 biomolecules-13-01447-f002:**
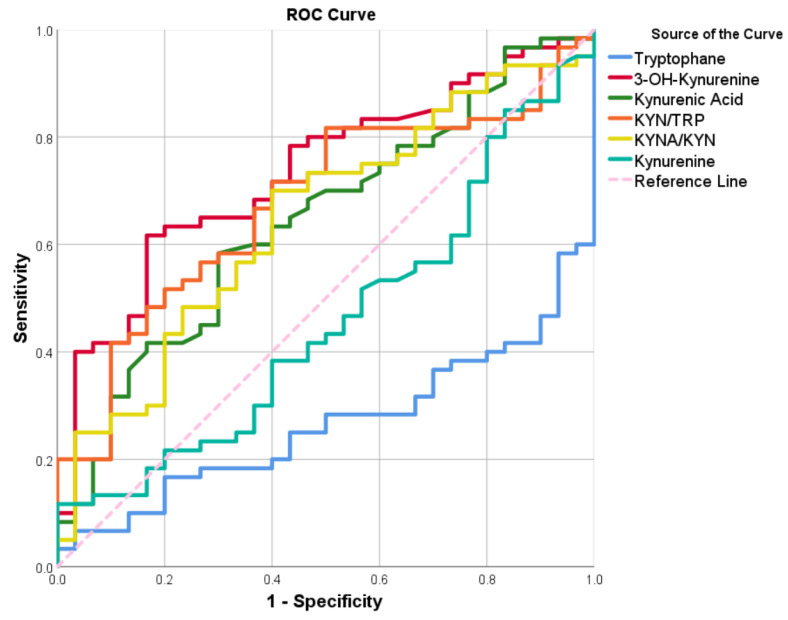
ROC of the TRYCATs parameters used for the diagnosis of PE.

**Table 1 biomolecules-13-01447-t001:** Demographic and clinical data in PE and control groups.

Variables	Control (*n* = 30)	PE (*n* = 60)	F/χ^2^	*p*
**Age years**	27.670 ± 5.933	29.230 ± 6.649	1.190	0.278
**Residency Urban/Rural**	13/17	27/33	0.023	0.881
**Previous abortion No/Yes**	22/8	30/30	4.464	**0.035**
**Nullipara No/Yes**	24/6	53/7	1.124	0.289
**Multipara** **No/Yes**	7/23	7/53	2.072	0.150
**Family history No/Yes**	29/1	40/20	10.857	**0.017**
**SBP mmHg**	122.200 ± 2.075	146.900 ± 17.445	59.389	**<0.001**
**DBP mmHg**	82.133 ± 2.030	89.033 ± 9.671	14.864	**<0.001**
**Gestational age Wks.**	30.370 ± 4.965	30.350 ± 3.0360	0.001	0.984
**Gravidity**	3.00 (1.00–4.00)	3.00 (1.25–4.75)	-	0.332
**Parity**	2.00 (0–3.00)	2.00 (0.25–3.75)	-	0.332
**Number of children**	2.00 (1.00–3.00)	3.00 (1.00–4.00)	-	0.092
**Cesarean delivery**	0(0–1)	1 (0.5–2.75)	-	**0.007**
**Natural delivery**	1.50 (0.75–3.00)	1.00 (1.00–3.00)	-	0.542

DBP: diastolic blood pressure, SBP: systolic blood pressure, F/χ^2^: F- or Chi-square statistic value, *p*: probability, Nullipara: no live births, Gravidity: number of pregnancies, Multipara: presence of life births, Parity: number of deliveries.

**Table 2 biomolecules-13-01447-t002:** Tryptophane catabolites in PE patients and control groups.

Variables	Control (*n* = 30)	PE (*n* = 60)	*p*
**TRP µM**	46.697 (43.534–53.376)	40.484 (36.743–47.486)	**<0.001**
**KYN µM**	1.629 (1.465–1.875)	1.689 (1.545–1.907)	0.502
**KYNA nM**	41.788 (35.985–45.096)	42.460 (39.191–51.290)	**0.027**
**3-HK nM**	30.464 (28.300–36.082)	39.814 (31.372–51.656)	**<0.001**
**KYNA/KYN**	25.025 (24.067–26.111)	24.706 (22.061–28.113)	**0.024**
**KYN/TRP**	0.034 (0.033–0.040)	0.042 (0.034–0.049)	**0.006**

TRP: tryptophane, KYN: kynurenine, KYNA: kynurenic acid, and 3-HK: 3-hydroxykynurenine. *p*: probability obtained by Mann-Whitney U test.

**Table 3 biomolecules-13-01447-t003:** Insulin resistance parameters in PE patients and control groups.

Variables	Control (*n* = 30)	PE (*n* = 60)	*p*
**Glucose mM**	5.241 ± 0.391	5.899 ± 0.914	**<0.001**
**Insulin pM**	52.097 (39.429–60.620)	65.439 (51.003–82.088)	**0.001**
**HOMA%B**	80.300 (72.675–100.150)	81.600 (67.925–102.150)	0.742
**HOMA%S**	99.750 (88.050–132.700)	82.750 (65.050–100.475)	**<0.001**
**HOMA2IR**	1.003 (0.754–1.136)	1.239 (0.995–1.537)	**<0.001**
**I/G**	9.663 (7.724–11.308)	11.177 (8.660–14.450)	**0.022**

*p*: probability obtained by Mann-Whitney U test except of glucose, which is computed by ANOVA as it is a normally distributed variable.

**Table 4 biomolecules-13-01447-t004:** Correlation matrix between the TRYCATs with the sociodemographic and clinical data.

Parameters	TRP	3-HK	KYNA	KYN	KYN/TRP	KYNA/KYN
**Residency**	−0.019	−0.089	0.087	0.022	0.002	0.046
**Abortion**	0.170	0.069	−0.072	0.005	−0.129	−0.026
**Nullipara**	−0.067	−0.153	−0.100	−0.085	0.033	−0.060
**Multipara**	0.024	0.173	0.100	0.044	−0.019	0.099
**Family history**	−0.171	0.177	−0.060	−0.124	0.060	0.051
**Age**	0.114	0.020	0.084	−0.013	−0.080	0.114
**SBP**	**−0.216 ***	**0.325 ****	0.125	−0.036	0.172	0.134
**DBP**	**−0.219 ***	0.182	0.093	−0.003	0.155	0.093
**Gestational age**	−0.081	−0.205	−0.114	0.003	0.071	−0.060
**Gravidity**	**0.261 ***	0.162	0.061	0.140	−0.172	−0.005
**having a child**	0.203	0.131	0.087	0.133	−0.132	0.000
**Cesarean delivery**	0.098	0.151	0.067	0.200	−0.016	0.011
**Natural delivery**	0.203	0.066	0.036	−0.052	**−0.227 ***	0.016
**Parity**	**0.261 ***	0.162	0.061	0.140	−0.172	−0.005
**Age of Onset**	**0.320 ***	−0.038	0.083	0.059	−0.229	0.111
**Duration of Symptoms**	−0.081	−0.105	−0.022	−0.077	−0.057	0.106
**Glucose**	−0.011	0.129	0.139	−0.039	0.041	0.143
**Insulin pM**	−0.205	0.096	0.005	−0.120	0.091	0.187
**HOMA%B**	−0.109	0.037	−0.153	−0.066	0.004	0.013
**HOMA%S**	0.191	−0.090	−0.034	0.129	−0.080	**−0.212 ***
**HOMA2IR**	−0.191	0.090	0.034	−0.129	0.080	**0.212 ***
**I/G**	−0.188	0.080	−0.074	−0.117	0.058	0.141

* *p* < 0.05, ** *p* < 0.01; statistical significance.

**Table 5 biomolecules-13-01447-t005:** Receiver operating characteristic-area under curve (AUC) analysis of the measured biomarkers used for diagnosis of PE.

Variables	Cut off	Sensitivity	Specificity	Youdin J Statistic	*p*	AUC	95% CI-AUC
**I/G nM**	10.189	65.0	64.3	0.29	**<0.001**	0.748	0.532–0.764
**HOMA2IR**	1.055	65.0	63.3	0.28	**<0.001**	0.736	0.630–0.841
**HOMA%S ***	94.25	63.3	63.3	0.27	**<0.001**	0.736	0.630–0.841
**Insulin pM**	54.62	63.3	63.3	0.27	**0.001**	0.721	0.613–0.828
**Glucose mM**	5.48	61.7	60.0	0.22	**0.001**	0.719	0.616–0.822
**HOMA%B**	80.6	50.0	50.0	0	0.742	0.479	0.358–0.599

*: The decrease of HOMA%S lower than the cut-off value predicts the presence of PE, CI: Confidence interval, *p*: the probability of seeing a sample AUC that is equal to or more than 0.5.

**Table 6 biomolecules-13-01447-t006:** ROC-AUC analysis of the measured biomarkers used for diagnosis of PE.

Variable(s)	Cut off	Sensitivity	Specificity	Youdin J Statistic	*p*	AUC	95% CI-AUC
**Tryptophane * µM**	44.896	66.7	68.3	0.35	**<0.001**	0.748	0.165–0.365
**3-HK** **nM**	38.491	65.0	66.7	0.32	**<0.001**	0.734	0.629–0.839
**KYN/TRP**	0.037	65.0	64.3	0.29	**0.006**	0.677	0.565–0.790
**KYNA/KYN**	24.935	60.0	60.0	0.20	**0.024**	0.646	0.528–0.764
**Kynurenic Acid nM**	41.591	60.0	60.0	0.20	**0.027**	0.643	0.525–0.762
**Kynurenine** **µM**	1.668	53.3	53.3	0.07	0.502	0.456	0.332–0.581

*: The decrease of tryptophane level lower than the cut-off value predicts the presence of PE, CI: Confidence interval, *p*: the probability of seeing a sample AUC that is equal to or more than 0.5.

## Data Availability

No new data were created or analyzed in this study. Data sharing does not apply to this article.

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
