# Peer review of "Maternal Tryptophan Catabolites and Insulin Resistance Parameters in Preeclampsia"

_biomolecules, 2023, doi:10.3390/biom13101447_

Round 1

Reviewer 1 Report

In the manuscript submitted by Jasim et al., the authors evaluated tryptophan metabolism, specifically, the kynurenine pathway, in preeclampsia (PE). Insulin resistance (IR) was also examined in PE patients. It was reported that tryptophan level was lower while the levels of kynurenine and related catabolites were increased in PE women, among whom fasting glucose and insulin were also higher.

As a serious condition for pregnant women, a better understanding of the pathogenesis of PE is important. Although the current study provided some interesting information, certain issues should be addressed:

1.       An increased kynurenine/tryptophan ratio has already been recognized in PE patients, therefore, the authors should downplay their claim in this, and should describe related researches in the Introduction. Also, in the Discussion the authors should discuss how their findings compared to others, e.g., why some are the same while others are different. In particular, in reference 39, “Kynurenic acid was elevated in the first trimester of pregnancy in women with PE. Furthermore, both kynurenic acid and picolinic acid were positively associated with proteinuria in women with PE in the third trimester of pregnancy. None of the other KP metabolites was changed in maternal blood, nor was any KP metabolite altered in umbilical cord blood.” The authors need to carefully compare their results with such statement and give reasonable explanation.

2.       In most studies chromatography was used to detect kynurenine and other catabolites, though ELISA is acceptable, the authors need to convert the concentration to mol/L for better comparison and validate their results were in the right range.

3.       Since the predictive efficacy of either kynurenine pathway or IR parameters was modest, it is suggested certain machine learning methods be applied for a better model construction.  

Minor language editing required.

Reviewer 2 Report

This paper aims to develop a prediction model for preeclampsia based on tryptophan and insulin resistance biomarkers.  While the rationale behind evaluating these factors is strong, the methods and presentation of results are insufficient to allow for publication of this paper in its current form.

Specifically:

1.     Prediction of PE is only helpful before the onset of disease, whereas these biomarkers were measured after the onset of PE.  In this way, reverse causation cannot be ruled out since the alterations observed could be a result of PE rather than a cause or predictor.  I would recommend NOT conducting prediction modeling, but rather, simply examine the association between these biomarkers and PE, controlling for confounders if possible due to the small N.  Authors should limit their conclusions to only what can be supported by this study design.

2.     The authors indicate in the Introduction that PE occurs in the third trimester.  While this is often the case, PE can occur anytime after 20 weeks, and thus, some cases occur in the second trimester.

3.     Introduction: The phrase “known to result” is too strong.  While there is a plethora of evidence suggesting that oxidative stress is indeed a part of the PE pathway, these pathways are connected by associations and not a causal connection is not firmly established.  Better to say “appear to result” or “are believed to result.”

4.     Methods: The authors report the age of PE cases (but not controls) in this section when this is, in fact, a result and should be moved to that section.

5.     Methods: Please explain the reason for overnight fasting as this is not obvious.

6.     Methods: Assuming that that was missing data (usually are), the authors should indicate how this was addressed.

7.     Unclear why ANOVA was used to analyze data since there are only two groups. 

8.     Details of how cut off values were determined for AUC analysis should be provided.  Were these from graphs, etc.?

9.     Results: No results include the effect size, but only the p-value, which is of limited utility.  All results in text and tables should be independent, such that a reader can either read the text or examine tables to determine findings.  It is poor practice to only include p-values as we cannot determine how big the effect is, which is more important for assessing clinical significance than the p-value.

10.  Table 1: Including Ns in the column headings suggests no missing data.  If there is missing data, there needs to be a column for the N for that variable. 

11.  Table 1: Reporting frequency by, for example, “13/17” makes it difficult for the reader to assess differences between groups.  These should be reported as count (frequency).

12.  Tables (in general): Report too many significant digits for the N included.  This implies more precision than they have.

13.  Tables (in general): Should include a footnote as to how the p-value was obtained for that table.

14.  Table 4: Testing to many correlations to not adjust for multiple comparisons.  Also, the vast majority of the correlations presented demonstrate weak correlation, which is not noted in results or discussion.

15.  ROC curves: The authors do not appear to correctly interpret the AUCs for several of the biomarkers (e.g., HOMA2 %B and HOMA2 %S) as the graph indicates that these predictors are worse than chance, not that they indicate a reduction. 

16.  Table 5: AUC for some factors (HOMA %S) do not make sense, given the graph.  The AUC is reported as 0.736 when it should be <0.5.

17.  Discussion: Line 214.  Not clear why the indicate reduced TRP as this showed a very small AUC, indicating poor prediction.  I do not see the result that they note.

18.  There is at least one study examining TRP as a predictor, but the authors do not cite it nor do they make comparisons to their results (PMID: 36569558).

19.  The mechanism in the discussion section should be tightened up as it is currently far too long.

20.  Discussion: Line 296. I cannot see where these results are presented in Table 5.

21.  Discussion: Line 311.  I disagree with these conclusions as the predictive ability (AUC) for all metabolites tested is below 0.8, indicating unacceptable predictive ability.

22.  Limitations: Line 333.  The most important limitation is the potential for reverse causality due to the selection of PE participants with diagnosed PE (and measurement of metabolites after the onset of disease).  Thus, one cannot rule out that these factors are a result of PE rather than having any real predictive potential.  Small sample size does not allow for modeling of many metabolites at once (or other clinical factors), which could improve the predictive ability.

Round 2

Reviewer 1 Report

No further questions.

Reviewer 2 Report

This revision does not address all concerns originially outlined by this reviewer, including the lack of detail regarding "correlations."  We are only told that they are significant, but no indication of the actual coefficient and direction of the effect is noted. Additionally, some of the numbers (AUCs) still don't make sense given the graph shown.  It is still unclear why ANOVA was used for two groups. I recommend rejection of this paper.
